# Effects of High-Intensity Ultrasound Pretreatment on the Exopolysaccharide Concentration and Biomass Increase in Cheese Whey Kefir

Ismael A. Encinas-Vazquez [1], Esther Carrillo-Pérez [1], Abraham R. Mártin-García [1], Carmen L. Del-Toro-Sánchez [2], Enrique Márquez-Ríos [2], Luis J. Bastarrachea [3,4] and José C. Rodríguez-Figueroa [1,*]

1   Chemical Engineering Department, University of Sonora, Hermosillo 83000, Mexico
2   Department of Food Science, University of Sonora, Hermosillo 83000, Mexico
3   Department of Nutrition, Dietetics and Food Science, Utah State University, 8700 Old Main Hill, Logan, UT 84322, USA; luis.bastarrachea@usu.edu
4   Department of Biological Engineering, Utah State University, 4105 Old Main Hill, Logan, UT 84322, USA
*   Correspondence: jose.rodriguez@unison.mx

**Abstract:** Cheese whey (CW) is the liquid by-product of cheese and yogurt making. This potential pollutant has high-quality nutrients exploitable through fermentation processes. Using high-intensity ultrasound on dairy products has shown several technological advantages for bioprocesses. Therefore, this study aimed to investigate the effect of high-intensity ultrasound (HIUS) on kefir grains biomass increase and specific metabolites in CW kefir. Fresh CW was ultrasonicated at $9.0 \pm 2.7$ and $18.0 \pm 3.0$ W/cm$^2$ for 30 and 180 s, inoculated with kefir grains, and fermented for 40 h. Total exopolysaccharide production, kefir grains biomass increase, titratable acidity, pH, and soluble solids were analyzed every 8 h. CW pretreated with $18.0 \pm 3.0$ W/cm$^2$ for 180 s and fermented for 16 h had significantly higher ($p < 0.05$) total exopolysaccharide concentration than the control: $212.7 \pm 0.0$ and $186.6 \pm 0.0$ mg/L, respectively. Ultrasonicated CW at 18 W/cm$^2$ for 30 and 180 s at 24 h fermentation time had significantly higher kefir grains biomass ($p < 0.05$) than the control: $44.2 \pm 0.8$ and $43.6 \pm 0.9$ g/L, and $40.5 \pm 0.4$ g/L, respectively. Fresh CW pretreated with HIUS enhanced the biosynthesis of kefir beverage total exopolysaccharides concentration and kefir grains biomass.

**Keywords:** biomass; cheese whey; high-intensity ultrasound; exopolysaccharide; kefir beverage; kefir grains

## 1. Introduction

Cheese whey (CW) is the liquid by-product from the precipitation of the hydrophobic milk casein fractions during cheese and yogurt making [1]. It has been reported that 1 kg of cheese is made with 10 L of milk, releasing approximately 9 L of CW [2]. CW global production has been estimated at 145 million tons per year [3], and approximately 50% is untreated and released directly into the environment [4]. Most of the CW is sweet whey, which contains 6.3% total solids, 4.9% lactose, 0.75% total protein, and 0.5% minerals ($w/w$) [5]. The presence of such organic matter generates high chemical oxygen demand (COD) and biological oxygen demand (BOD): 50–102 g/L and 27–60 g/L, respectively. Therefore, CW is considered the most critical pollutant in dairy wastewater [6]. However, the chemical composition of CW includes valuable essential nutrients that can be used as substrates in a wide variety of bioprocesses.

Kefir is a traditional fermented milk popular worldwide. It is a functional food with health-promoting effects on rectal colon cancer, cardiovascular diseases, type 2 diabetes mellitus, obesity, and kidney diseases. It also has positive effects on the modulation of the immune system, and intestinal microbiota. This beverage can be produced by industrial

or artisanal methods. Milk is inoculated by kefir grains when the artisanal method is applied, so the amount of inoculum is a limiting factor [7]. Kefir grains biomass mainly comprises microorganisms, water, proteins, and polysaccharides [8]. Kefiran is an extracellular polysaccharide metabolite present in kefir grains. This biodegradable and edible biopolymer has several relevant applications, such as an encapsulating agent for drug delivery [9]. Biomass produced by microorganisms and vegetables usually presents nontoxicity and versatility, so it has been evaluated in recent research as a raw material through industrial encapsulating processes [9,10]. Moreover, kefiran has been associated with many bioactivities due to its antimicrobial, antioxidant, antiproliferative, antitumor, and prebiotic properties [9,11,12]. Recently, a metagenomic sequencing analysis revealed the presence of 522 species in the microbiota of this multi-functional beverage. An abundance of metabolites and metabolic pathways during incubation was also found. The chemical structure of the available nutrients in the media influence kefir metabolites biosynthesis [13]. Therefore, it is hypothesized that the application of emergent technologies to the kefir media may impact its metabolism.

New reviews have summarized the effects of ultrasound vibration, streaming, and cavitation on dairy products [14]. Ultrasound has been regarded as environmentally friendly, low cost, and easy to maintain and operate [15]. High-intensity ultrasonic processing substrates of fermented dairy products have enhanced product stability, reduced processing time, and improved quality [16].

To the best of our knowledge, kefir production using CW as a substrate has scarcely been studied. Lactose consumption, ethanol production, kefir biomass increase, volatile compounds, and organic acids formation have been reported [17,18]. However, the implications of high-intensity ultrasound pretreatment on CW followed by fermentation using kefir grains have yet to be explored. In fact, it has been hypothesized that CW pretreated with high-intensity ultrasound may impact kefir grains microorganisms' metabolism, increasing metabolite synthesis with relevant health benefits. Therefore, this research aimed to investigate the effect of high-intensity ultrasound (HIUS) as a pretreatment on the exopolysaccharide concentration and biomass increase in CW kefir.

## 2. Materials and Methods

Experimental flow diagram (Figure 1) is as follows:

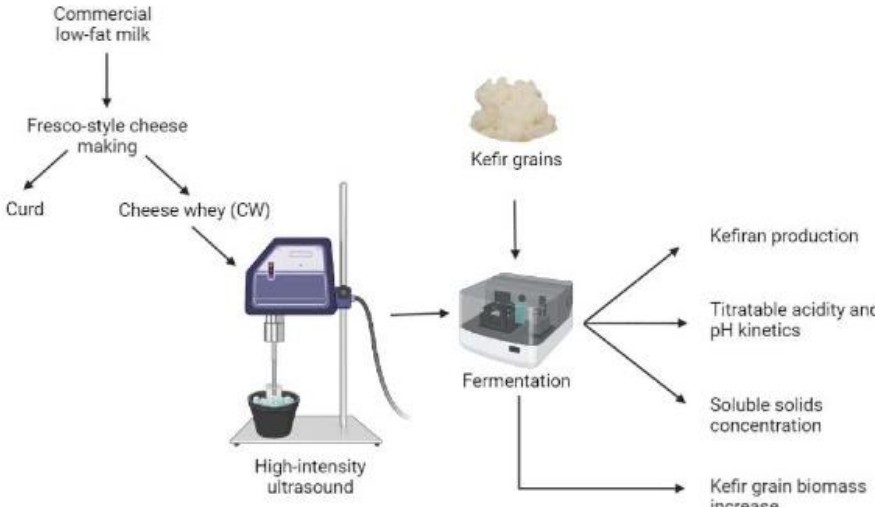

**Figure 1.** Experimental flow diagram to study the effect of HIUS as pretreatment on CW fermented by kefir grains [19].

### 2.1. CW Production

CW was obtained as a by-product of Fresco-style cheese production with some modifications [20]. Commercial pasteurized low-fat milk (50 g/L carbohydrates, 30 g/L pro-

tein, and 10 g/L fat) was heated at 32–36 °C. Commercial rennet (0.15 mL/L) and CaCl$_2$ (0.83 g/L) were added and set undisturbed for 1.5 h, approximately. Once the curd was formed, it was cut with a knife, cooked at 40 °C for 25–30 min, set undisturbed for 30 min, and filtered through a cheesecloth to obtain fresh CW. This last step was repeated twice.

## 2.2. Kefir Grains Activation

Inactivated kefir grains were obtained from the culture collection of the Laboratory of Food Engineering at the University of Sonora (Hermosillo, Mexico) [21]. They were inoculated into commercial UHT skimmed cow's milk (48 g/L carbohydrates, 31.2 g/L protein, and 6 g/L fat) and incubated at 25 ± 3 °C for 24 h without stirring. Kefir grains were retrieved with plastic sieving and rinsed with distilled water. This procedure was repeated for 3 weeks [22].

## 2.3. Treatments

CW samples were cooled at 6 °C and pretreated with high-intensity ultrasound (HIUS) in 600 mL aliquots placed in a glass vessel while cooled to keep its temperature at <12 °C. Treatments are shown in Table 1. Sonication was conducted by introducing an ultrasonic horn (22 mm diameter and 100 mm length, S24d22D, Hielscher Ultrasonic, Germany) with 400 W ultrasound equipment (UP400St, Hielscher Ultrasonic, Germany) at 70% initial amplitude and 23.0 ± 0.9 kHz frequency, 3 cm deep into the CW solution. The actual intensities delivered to CW samples were calculated by the calorimetric method [23], resulting 9.0 ± 2.7 and 18 ± 3.0 W/cm$^2$. HIUS treated CW was inoculated in Erlenmeyer flasks with 3% (*w/v*) kefir grains, capped with a porous cloth to let gases interchange and protect fermentation, incubated at 25 ± 1 °C for 40 h at 100 rpm agitation rate (CVP-500, CScientific, USA) [21]. Samples were analyzed every 8 h during fermentation. The experiment was repeated three times.

**Table 1.** HIUS cheese whey (CW) kefir treatments.

| Treatments (T) | Kefir Fermentation | HIUS [1] (W/cm$^2$) | HIUS [1] Time (s) |
|:---:|:---:|:---:|:---:|
| T0 | - | - | - |
| T1 | + | - | - |
| T2 | + | 9.0 ± 2.7 | 30 |
| T3 | + | 9.0 ± 2.7 | 180 |
| T4 | + | 18.0 ± 3.0 | 30 |
| T5 | + | 18.0 ± 3.0 | 180 |

[1] High-intensity ultrasound pretreatment.

## 2.4. Exopolysaccharide Quantification

HIUS CW kefir exopolysaccharide was quantified using 3 mL aliquots, heated in a boiling water bath for 15 min, and cooled at room temperature. Samples were vortexed (Vxmnal, Parsippany, NJ, USA) and centrifuged (Sorvall Lynx 4000, Thermo Scientific, Germany) at 10,000× *g* for 15 min at 20 °C. Two volumes of cold ethanol were added to the supernatant, maintained at −20 °C for 24 h, and centrifuged at 10,000× *g* for 15 min at 4 °C. Then, the pellets were dissolved in hot water and precipitated again with two volumes of cold ethanol, maintained at −20 °C for 24 h, and centrifuged at 10,000× *g* for 15 min at 4 °C. The pellets containing exopolysaccharides were resuspended in distilled hot water [24,25]. Total sugars were analyzed through the colorimetric anthrone method at 620 nm [26]. Glucose absorbance (0–200 mg/L) was used as the standard to obtain the calibration curve.

## 2.5. Kefir Grains Biomass Increase

Kefir grains biomass was separated from the kefir beverage through a plastic sieve. The biomass was rinsed with tap water, left for 5 min, and weighted using an analytical balance (ZSA120, Scientech, Boulder, CO, USA). Biomass increase was calculated using

the gravimetric method [27]. Samples were obtained at 0, 8, 16, 24, 32, and 40 h fermentation time.

### 2.6. Physicochemical Analysis

Kefir beverages samples were analyzed through titratable acidity, pH, and soluble solids concentration (°Brix) during ultrasonicated CW fermentation time. Titratable acidity, pH, and °Brix were measured using 947.05, 981.12, and 932.14 AOAC International methods, respectively [28].

### 2.7. Statistical Analysis

All experiments were repeated three times. Samples were analyzed in triplicate, and results expressed the mean value ± standard deviation (SD). A generalized linear model (GLM) analysis of variance (ANOVA) was used to find out differences between means at $\alpha$ = 0.05 significant level. Significant differences between means were analyzed by the Tukey–Kramer test ($p < 0.05$). All statistical analysis was performed using the NCSS 2012 statistical software (NCSS Inc., Kaysville, UT, USA).

## 3. Results and Discussion

### 3.1. High Intensity Ultrasound Effects on Total Exopolysaccharides Production

CW pretreated with HIUS increased the amount of total exopolysaccharide in kefir beverages (Figure 2). Results showed the presence of exopolysaccharides during the fermentation time. Kefiran is a water-soluble exopolysaccharide; therefore, once kefir grains were inoculated, it was quantified from 108.5 ± 0.0 to 136.3 ± 0.0 mg/L at 0 h fermentation time. CW pretreated with 18.0 ± 3.0 W/cm$^2$ HIUS for 180 s and fermented for 16 h by kefir grains microorganisms had significantly higher ($p < 0.05$) total exopolysaccharide concentration than control: 212.7 ± 0.0 and 186.6 ± 0.0 mg/L, respectively. In a previous study, the exopolysaccharide concentration of commercial brands of kefir beverages in Russia was investigated. They evaluated seven different kefir beverage brands presenting 50.9 ± 8.1 to 202.5 ± 19.4 mg/L total exopolysaccharide [29]. Exopolysaccharide production depends on the growth conditions and the medium chemical composition, especially the absolute quantities, sources, and ratio of carbon/nitrogen [30]. The fermentation of HIUS milk substrate has been associated with increased rates of the reduction in lactose and the production of glucose, galactose, and oligosaccharides with a degree of polymerization of the three [31].

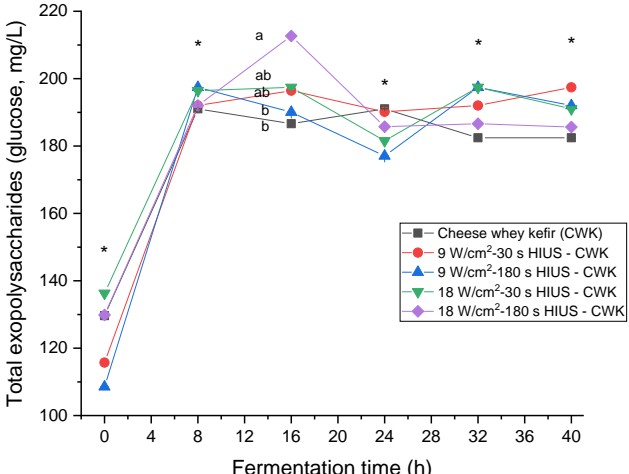

**Figure 2.** Total exopolysaccharides production in CW pretreated with HIUS by kefir grains during fermentation. * Treatments in the corresponding fermentation time are not significantly different ($p > 0.05$). Different lowercase letters indicate significant differences ($p < 0.05$). HIUS = High-intensity ultrasound pretreatment.

Moreover, these effects increase with the HIUS time extension [32]. In addition, the application of HIUS on CW proteins has demonstrated a decrease in particle size, narrowing of their distribution, increase in the specific free surface, partial cleavage of intermolecular hydrophobic interactions, and a significant decrease in molecular weight and protein fractionation [33]. These facts increase the amount and availability of substrates for enzymatic activity and may boost the metabolic activity of kefir grains microorganisms to biosynthesize exopolysaccharides. In a previous study, non-fat reconstituted milk was pretreated with HIUS at $22 \pm 1.25$ kHz, 20 W/cm$^2$, and 90 W/L for 3 min and fermented during 8–12 h with kefir grains. The highest exopolysaccharide amount found in the kefir beverage was $204 \pm 0.94$ mg/L. It was also found that 18% more exopolysaccharide was produced in ultrasonicated treatments than in controls [34]. Table 2 shows the increasing percentage of total exopolysaccharide production of all treatments and the control as a reference with its content at 0 h fermentation time. The maximum exopolysaccharides increase was found in CW ultrasonicated at $9.0 \pm 2.7$ W/cm$^2$ for 3 min and fermented for 8 and 32 h. The total exopolysaccharides increase found was 35% higher than the control.

**Table 2.** Total exopolysaccharides increase during fermentation time (%).

| Fermentation Time (h) | Total Exopolysaccharide Increase (%) | | | | |
| | HIUS CW Treatments | | | | |
| | T1 | T2 | T3 | T4 | T5 |
|---|---|---|---|---|---|
| 8 | $47.3 \pm 6.7$ cA | $65.9 \pm 8.1$ bA | $81.9 \pm 0.0$ aA | $44.1 \pm 1.3$ cA | $47.9 \pm 7.2$ cAB |
| 16 | $43.9 \pm 7.2$ bA | $69.8 \pm 1.5$ aA | $75.1 \pm 7.0$ aAB | $44.8 \pm 0.0$ bA | $63.7 \pm 10.2$ aA |
| 24 | $47.3 \pm 6.7$ abA | $64.3 \pm 6.5$ aA | $63.1 \pm 6.7$ aB | $33.2 \pm 9.5$ bA | $43 \pm 5.8$ bB |
| 32 | $40.7 \pm 11.1$ bA | $65.9 \pm 8.1$ aA | $81.9 \pm 0.0$ aA | $44.8 \pm 0.0$ bA | $43.7 \pm 7.2$ bB |
| 40 | $40.7 \pm 11.1$ bA | $70.6 \pm 0.0$ aA | $76.9 \pm 8.6$ aAB | $40.1 \pm 6.3$ bA | $42.9 \pm 5.9$ bB |

T1 = Cheese whey kefir (CWK), T2 = 9 W/cm$^2$-30 s CWK, T3 = 9 W/cm$^2$-180 s CWK, T4 = 18 W/cm$^2$-30 s CWK, T5 = 18 W/cm$^2$-180 s CWK. Data represent the mean values $\pm$ standard deviation (n = 3). Different capital letters within a column indicate significant differences ($p < 0.05$). Different lowercase letters within a row indicate significant differences ($p < 0.05$).

Total exopolysaccharides production was investigated during fermentation time. Potoroko et al. [34] reported that HIUS pretreatment on reconstituted milk generated exopolysaccharide biosynthesis at the first hours of fermentation time, and it had its peak at 24–60 h in the stationary phase. Cheirsilp and Radchabut [35] found that kefiran production began in the early exponential growth phase when using a mixed culture of *Lactobacillus kefiranofaciens* JCM 6985 and *Saccharomyces cerevisiae* IFO 0216. In this study, kefiran biosynthesis also started during the logarithmic growth phase, and it had its maximum peak in the same growth phase at 16 h fermentation time. Therefore, HIUS may have modified the bioavailability of nutrients' structural and chemical composition, stimulating the metabolic activity to accelerate exopolysaccharide biosynthesis.

### 3.2. Kefir Grain Biomass Increase

The application of HIUS on CW improved kefir grains biomass concentration. Figure 3 shows the effect of HIUS on kefir grains biomass increase during fermentation time. Results showed that most of the HIUS pretreatments had higher kefir grains biomass than the control during the kinetic of fermentation. HIUS pretreatments at 18 W/cm$^2$ for 30 and 180 s at 24 h fermentation time had significantly ($p < 0.05$) higher kefir grains biomass than control: $44.2 \pm 0.8$ and $43.6 \pm 0.9$ g/L, and $40.5 \pm 0.4$ g/L, respectively. Even though there was not a significant difference ($p > 0.05$) at 32 h fermentation time, control and HIUS pretreatment at 18 W/cm$^2$ for 30 s presented the highest values: $46.9 \pm 0.5$ and $46.9 \pm 1.6$ g/L, respectively. It has been reported that UHT milk at 80 rpm agitation rate and organic skim milk at 125 rpm rotation rate fermented for 24 h at 25 °C had 18.8 g/L and 38.9 g/L kefir grains biomass increase, respectively [36,37]. They increased their mass due to the growth of microorganisms and the biosynthesis of their chemical components, such as proteins and exopolysaccharides. Kefiran plays an essential role in the formation

and stabilization of these grains [38]. Wang et al. [38] proposed a model of the mechanism of kefir grains formation using lactic acid bacteria and yeasts isolates. They found that the whole outer layer of the grain had more microorganisms than the inner part. These microorganisms were *lactobacilli*, *lactococci*, and yeasts. They reported their abilities to auto-aggregate, co-aggregate, form biofilm, and properties, such as hydrophobicity and cell surface. Therefore, kefir lactic acid bacteria and yeast cells density increases, and chemical milk components in the liquid phase accumulate on the granule surface to form novel kefir grains. Wang et al. [38] also found that kefir grains biomass increased significantly from 1 to 3 h of fermentation time. In the present study, an atypical growth curve was observed during fermentation time. Kefir grains biomass had the maximum increase at 8 h incubation time, decreased at 16 h, and increased until 32 h (Figure 3). This decrease in kefir grains biomass coincided with the maximum increase in total exopolysaccharide concentration found in CW kefir beverage pretreated with HIUS at 18 W/cm$^2$ for 180 s. In fact, the correlation coefficient (R$^2$) between those parameters was R$^2$ = 0.50. These findings suggest that the mechanism to increase kefir grain biomass could have a cyclic behavior with the migration of chemical components from the grains into the medium and finally the deposition onto them during fermentation time. These findings may be an example of the model mechanism proposed by Wang et al. [38].

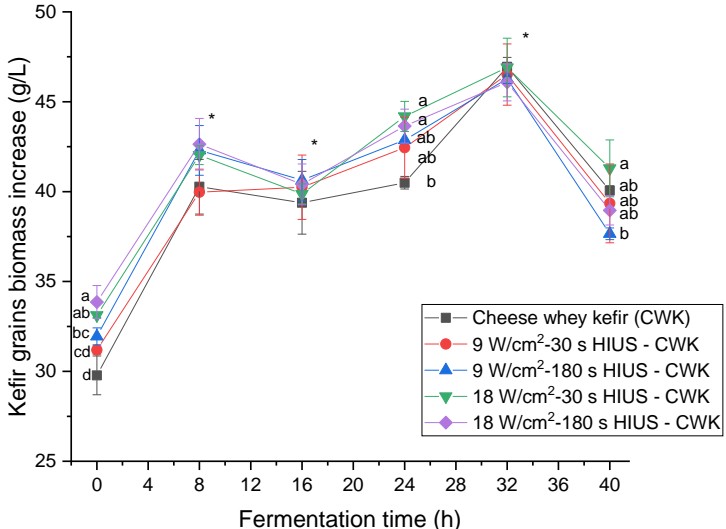

**Figure 3.** Kefir grains biomass increase in CW pretreated with HIUS during fermentation. * Treatments in the corresponding fermentation time are not significantly different ($p > 0.05$). Different lowercase letters indicate significant differences ($p < 0.05$). HIUS = High-intensity ultrasound pretreatment.

Kefir grains biomass percentage increase during fermentation was also analyzed (Table 3). The initial kefir grains inoculum was 30 g/L. The highest biomass increase values were 53.6–56.4% at 32 h fermentation. At this fermentation time, a significant ($p > 0.05$) difference was not found between the control and CW ultrasonicated treatments. Previous studies reported that 60 g/L deproteinized CW lactose substrate fermented for 120 h at 25 °C (without agitation), and low-fat cow's milk substrate fermented for 24 h at 25 °C (100 rpm agitation rate), had 43.0% and 32% kefir grains biomass increase [22,39].

**Table 3.** Kefir grains biomass increase during fermentation time (%).

| Fermentation Time (h) | Kefir grains Biomass Increase (%) | | | | |
|---|---|---|---|---|---|
| | HIUS CW Treatments | | | | |
| | T1 | T2 | T3 | T4 | T5 |
| 0 | 1.3 ± 0.5 cC | 3.9 ± 0.8 cC | 6.4 ± 1.6 bE | 10.4 ± 0.4 abD | 12.8 ± 3.1 aE |
| 8 | 34.2 ± 5.1 aB | 33.2 ± 4.3 aB | 40.9 ± 4.6 aBC | 40.0 ± 1.7 aBC | 42.1 ± 4.8 aBC |
| 16 | 31.2 ± 5.8 aB | 34.1 ± 6.0 aB | 35.4 ± 3.8 aC | 32.9 ± 1.9 aC | 34.6 ± 3.8 aCD |
| 24 | 34.9 ± 1.2 cB | 41.4 ± 5.6b cAB | 42.8 ± 0.1 bcB | 47.2 ± 2.8 abAB | 45.4 ± 3.1 abAB |
| 32 | 56.4 ± 1.7 aA | 55.0 ± 5.7 aA | 54.2 ± 0.9 aA | 56.3 ± 5.4 aA | 53.6 ± 3.4 aA |
| 40 | 33.5 ± 0.9 abB | 31.1 ± 7.3 abB | 25.5 ± 1.1 bD | 37.6 ± 5.3 aC | 29.8 ± 2.7 abD |

T1 = Cheese whey kefir (CWK), T2 = 9 W/cm$^2$-30 s CWK, T3 = 9 W/cm$^2$-180 s CWK, T4 = 18 W/cm$^2$-30 s CWK, T5= 18 W/cm$^2$-180 s CWK. Data represent the mean values ± standard deviation (n = 3). Different capital letters within a column indicate significant differences ($p < 0.05$). Different lowercase letters within a row indicate significant differences ($p < 0.05$).

### 3.3. Titratable Acidity and pH Kinetics

Titratable acidity increased during kefir fermentation time. Figure 4 shows the titratable acidity kinetics. CW had 0.13 ± 0.00% acidity initially. At 40 h fermentation time, HIUS CW kefir had 1.53–1.59% acidity. There was no statistically significant difference ($p > 0.05$) between treatments and CW kefir at this time. However, at 24 h of fermentation, the titratable acidity values of treatments T4, T3, and T5 were significantly different ($p < 0.05$) from CW kefir; 1.08 ± 0.01%, 1.07 ± 0.02%, 1.06 ± 0.03%, and 1.01 ± 0.01%, respectively. Motaghi et al. [40] evaluated the fermentation of cow's milk with kefir grains at 25 °C for 24 h, without agitation. They reported a 1.47% titratable acidity value. Treatments T4, T3, and T5 also presented the highest kefir grains biomass concentrations at the same fermentation time, 44.2 ± 0.8 g/L, 43.6 ± 0.9 g/L, and 42.9 ± 0.0 g/L, respectively. Therefore, the correlation coefficient between these two parameters was analyzed. The coefficient of correlation of treatments T4, T3, and T5 was R$^2$ = 0.87, R$^2$ = 0.83, and R$^2$ = 0.79, respectively.

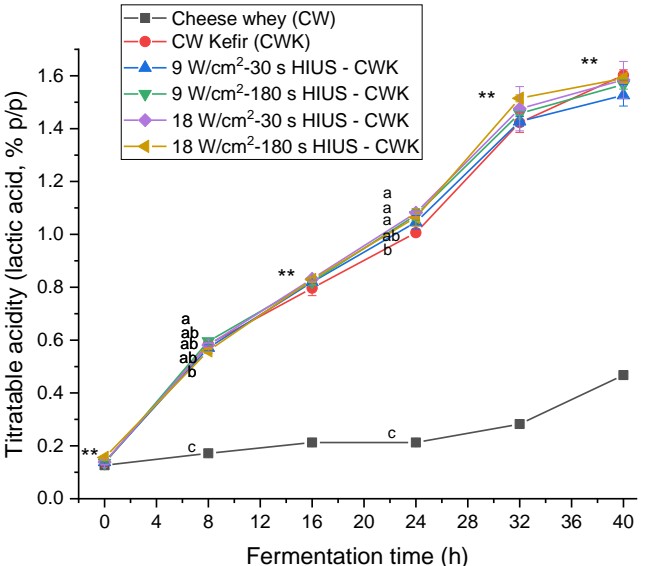

**Figure 4.** Titratable acidity concentration in CW pretreated with HIUS and fermented by kefir grains. ** Treatments in the corresponding fermentation time are not significantly different ($p > 0.05$). Different lowercase letters indicate significant differences ($p < 0.05$). HIUS = High-intensity ultrasound pretreatment.

All the HIUS CW treatments showed a decrease in pH through fermentation time. Results are shown in Figure 5. Initially, CW had a pH of 6.48; at the end of the fermentation, all treatments reached a pH of 3.3 pH. The treatments' pH was not significantly different ($p > 0.05$) from CW kefir during fermentation time. It has been reported, that during the first 24 h of incubation, homofermentative lactic acid *streptococci* grow rapidly, causing

a drop in pH [8]. Zajšek and Goršek [41] reported a 4.22 ± 0.04 pH value in HTP milk inoculated by kefir grains and incubated at 24 °C for 24 h. In this study, the highest pH decrease was observed at 8 h fermentation time, reaching a value of 3.9. After 24 h, all treatments had a pH value of 3.5. A previous study established that pH stabilizes at 3.3 [37]. This pH value was reached and stabilized by all treatments after 32 h of fermentation. The coefficient of correlation between kefir grains biomass and pH was also analyzed in this study, considering treatments with the highest biomass increase. The coefficient of correlation of treatments T4, T3, and T5 was, $R^2 = 0.85$, $R^2 = 0.92$, and $R^2 = 0.89$, respectively.

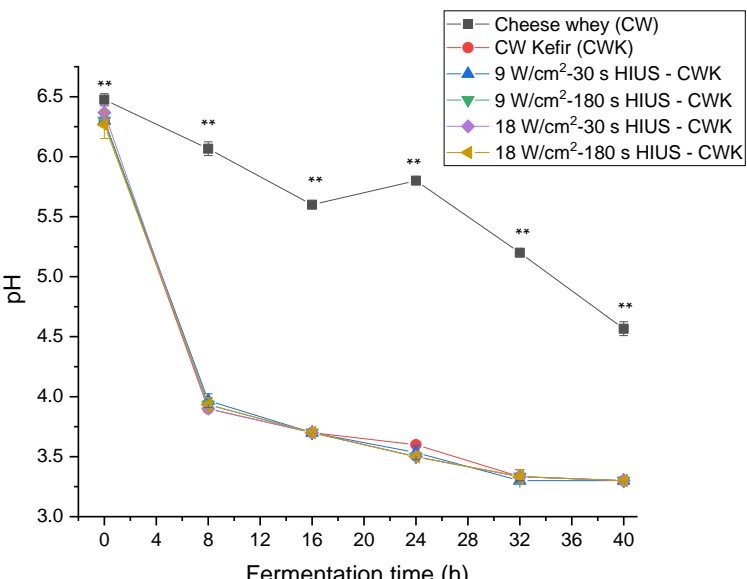

**Figure 5.** Changes of pH in CW pretreated with HIUS and fermented by kefir grains. ** Treatments in the corresponding fermentation time are not significantly different ($p > 0.05$). Different lowercase letters indicate significant differences ($p < 0.05$). HIUS = High-intensity ultrasound pretreatment.

### 3.4. Soluble Solids Concentration

The effect of high-intensity ultrasound on CW kefir beverages' soluble solids concentration was evaluated. HIUS CW kefir decreased soluble solids concentration (° Brix) at the end of fermentation time (Figure 6). CW initially had 6.45 ± 0.06 °Brix, whereas treatments had a 5.5–5.9 °Brix. Soluble solids concentration of the treatments was not statistically different ($p > 0.05$) at 40 h of fermentation. °Brix increased and decreased during fermentation time (Table 4). It was hypothesized that soluble solids concentration decreased all the time as a consequence of the use of CW nutrients as part of the kefir grains microbial metabolism. However, Table 4 data showed that kefir grains biomass increased, and at the same time, soluble solids concentration decreased at 8, 16, 24, and 32 h of fermentation. Therefore, these results suggest an inverse relationship between these two parameters. This kefir grains biomass and soluble solids concentration behavior may be explained by the mechanism of kefir grains formation during fermentation proposed by Wang et al. [38].

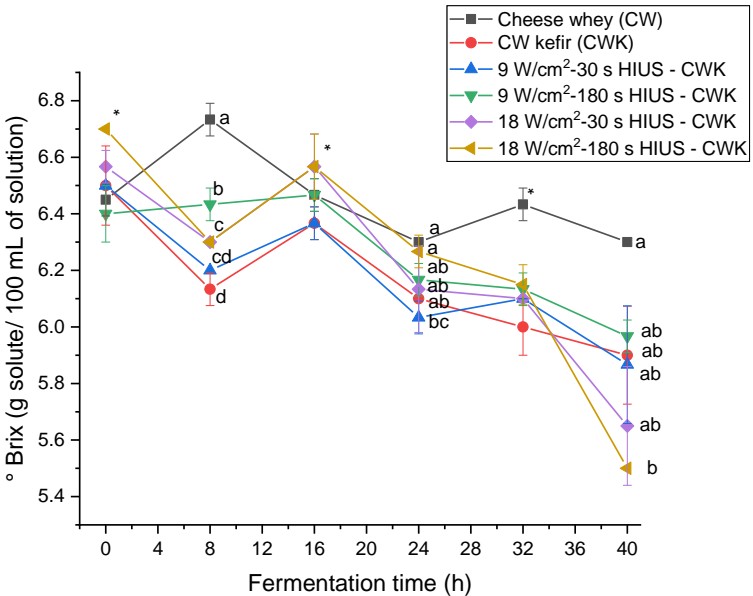

**Figure 6.** Soluble solids concentration in CW pretreated with HIUS by kefir grains during fermentation time. * Treatments in the corresponding fermentation time are not significantly different ($p > 0.05$). Different lowercase letters indicate significant differences ($p < 0.05$). HIUS = High-intensity ultrasound pretreatment.

**Table 4.** Changes in the parameters of all CW HIUS pretreated kefir during fermentation.

| | Parameters Change during Cheese Whey Kefir Fermentation | | | |
|---|---|---|---|---|
| | **Fermentation Time (h)** | | | |
| **Parameter** | **8** | **16** | **24** | **32** |
| Kefir grains biomass (g/L) | ✛ 11.7 | ▬ 1.4 | ✛ 3.0 | ✛ 3.2 |
| Total exopolysaccharide (glucose, mg/L) | ✛ 71.9 | ✛ 4.7 | ▬ 15.5 | ✛ 9.8 |
| Soluble solids (g solut/100 mL solution) | ▬ 0.23 | ✛ 0.18 | ▬ 0.34 | ▬ 0.03 |

Data = all treatments' mean values compared with the last value obtained before ✛ = the parameter concentration increased compared with the value obtained before. ▬ = the parameter concentration decreased compared with the value obtained before.

## 4. Conclusions

This research aimed to explore the effect of high-intensity ultrasound on kefir grains biomass increase and total exopolysaccharides concentration in CW kefir. CW pretreated with $18.0 \pm 3.0$ W/cm$^2$ HIUS for 180 s improved total exopolysaccharides concentration significantly. The highest kefir grains biomass increase was also presented by this HIUS treatment for 30 and 180 s. No significant HIUS effects were observed on pH and titratable acidity at the end of the fermentation. However, these two parameters were correlated with kefir grains biomass increase. An apparent HIUS effect on soluble solids concentration was not found. Conversely, an inverse relationship between kefir grains biomass increase and soluble solids concentration was observed.

Finally, this study demonstrated that applying HIUS to fresh CW enhances the biosynthesis of kefir grains biomass and health-related metabolites by the kefir microorganisms. This study showed that using this emergent technology may reduce the potential negative effects of CW in the environment. Sensory analysis and metabolomic studies are needed to develop a CW kefir accepted by consumers and to understand the HIUS effect on the metabolic pathways during fermentation time.

**Author Contributions:** Conceptualization, E.C.-P., A.R.M.-G., C.L.D.-T.-S. and J.C.R.-F.; methodology, I.A.E.-V. and J.C.R.-F.; validation, E.M.-R.; investigation, I.A.E.-V. and J.C.R.-F.; resources, I.A.E.-V. and J.C.R.-F.; writing—original draft preparation, I.A.E.-V. and J.C.R.-F.; writing—review and editing, E.C.-P., A.R.M.-G., C.L.D.-T.-S., E.M.-R., L.J.B. and J.C.R.-F.; supervision, E.C.-P., A.R.M.-G., C.L.D.-T.-S., L.J.B. and J.C.R.-F.; project administration, J.C.R.-F.; funding acquisition, J.C.R.-F. All authors have read and agreed to the published version of the manuscript.

**Funding:** Authors would like to thank the Mexican Science and Technology Council (CONACYT) and the Secretary of Public Education for kindly funding this research project, through the program CB 2015-258483-Z and a bachelor's degree scholarship received by Ismael Antonio Encinas-Vázquez.

**Data Availability Statement:** If there is interest in the data, please contact the corresponding author.

**Acknowledgments:** Authors would like to thank César Otero León for the technical support given to this project.

**Conflicts of Interest:** The authors declare no conflict of interest.

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
