# Peer review of "Effects of High-Intensity Ultrasound Pretreatment on the Exopolysaccharide Concentration and Biomass Increase in Cheese Whey Kefir"

_processes, doi:10.3390/pr11071905_

Round 1

Reviewer 1 Report

1.      As your abstract's final sentence, include a "take-home" message.

2.      Put the keywords in a new order based on alphabetical order.

3.      In the present form, nothing really novel. The current study appears to be a replication or modified study according to the lack of novelty. The authors must extensively describe the novel their work is. This work should be rejected due to a serious concern.

4.      In order to highlight the gaps in the literature that the most recent research aims to fill, it is crucial to review the benefits, novelty, and limitations of earlier studies in the introduction.

5.      In the last paragraph of the introduction, please explain the objective of the present article.

6.      Studies related biomass needs to highlight, one of the would be from Santoso et al. as follow: Power and Energy Optimization of Carbon Based Lithium-Ion Battery from Water Spinach (Ipomoea Aquatica). J. Ecol. Eng. 2023, 24, 213–23. https://doi.org/10.12911/22998993/158564

7.      To help the reader grasp the study's workflow more easily, the authors could include more visuals to the materials and methods section in the form of figures rather than sticking with the text that now predominates.

8.      Additional information about tools used, such as the maker, country, and specification, should be included.

9.      The revised manuscript after peer review must provide detailed information on the error and tolerance of the experimental equipment utilized in this study. Due to the disparate outcomes of other researchers' subsequent studies, it would make for a valuable discussion.

10.   Results comparison with similar previous studies needs to give.

11.   The authors need to improve the discussion in the present article become more comprehensive. The present form was insufficient.

12.   Please include the limitation of the present study, it is missing.

13.   Mention further research in the conclusion section.

14.   The authors should enrich their references in the revised manuscript from five years ago. MDPI reference is strongly recommended.

15.   The manuscript needs to be proofread by the authors since it has grammatical and language issues.

16.   After revision, provide a graphical abstract for submission.

Author Response

Dear Prof. reviewer, thank you so much for your comments.

These are our responses:

Point 1. As your abstract's final sentence, include a "take-home" message.     

Response 1:              

In fact, it has been hypothesized that CW pretreated with high-intensity ultrasound may impact kefir grains microorganisms´ metabolism, increasing metabolite synthesis with relevant health benefits. Therefore, this research aimed to investigate the effect of high-intensity ultrasound as pretreatment on the exopolysaccharide concentration and biomass increase of CW kefir

Point 2. Put the keywords in a new order based on alphabetical order.

Response 2:                          

Keywords: biomass; cheese whey; high-intensity ultrasound; kefiran; kefir beverage; kefir grains

Point 3. In the present form, nothing really novel. The current study appears to be a replication or modified study according to the lack of novelty. The authors must extensively describe the novel their work is. This work should be rejected due to a serious concern.

Response 3:              

Kefir production using CW as substrate has scarcely been studied. Lactose consumption, ethanol production, kefir biomass increase, volatile compounds and organic acids formation have been reported [14,15]. However, the implications of high-intensity ultrasound pretreatment followed by the fermentation using kefir grains on CW have not been explored. In fact, it has been hypothesized that CW pretreated with high-intensity ultrasound may impact kefir grains microorganisms´ metabolism, increasing metabolite synthesis with relevant health benefits. Therefore, this research aimed to investigate the effect of high-intensity ultrasound as pretreatment on the exopolysaccharide concentration and biomass increase of CW kefir.

Point 4. In order to highlight the gaps in the literature that the most recent research aims to fill, it is crucial to review the benefits, novelty, and limitations of earlier studies in the introduction.

Response 4:              

Kefir production using CW as substrate has scarcely been studied. Lactose consumption, ethanol production, kefir biomass increase, volatile compounds and organic acids formation have been reported [14,15]. However, the implications of high-intensity ultrasound pretreatment followed by the fermentation using kefir grains on CW have not been explored. In fact, it has been hypothesized that CW pretreated with high-intensity ultrasound may impact kefir grains microorganisms´ metabolism, increasing metabolite synthesis with relevant health benefits. Therefore, this research aimed to investigate the effect of high-intensity ultrasound as pretreatment on the exopolysaccharide concentration and biomass increase of CW kefir.

Point 5. In the last paragraph of the introduction, please explain the objective of the present article

Response 5:              

This research aimed to investigate the effect of high-intensity ultrasound as pretreatment on the exopolysaccharide concentration and biomass increase of CW kefir.

Point 6. Studies related biomass needs to highlight, one of the would be from Santoso et al. as follow: Power and Energy Optimization of Carbon Based Lithium-Ion Battery from Water Spinach (Ipomoea Aquatica). J. Ecol. Eng. 2023, 24, 213–23. https://doi.org/10.12911/22998993/158564

Response 6:              

Microorganisms and vegetable biomass present nontoxicity and versatility, so novel research has considered them as raw material through industrial processes [9,10].

  1. Moradi, Z.; Kalanpour, N. Kefiran, a Branched Polysaccharide: Preparation, Properties and Applications: A Review. Carbohydr Polym 2019, 223, doi:10.1016/j.carbpol.2019.115100.
  2. Santoso, B.; Ammarullah, M.; Haryati, S.; Sofijan, A.; Bustan, M.D. Power and Energy Optimization of Carbon Based Lithium-Ion Battery from Water Spinach (Ipomoea Aquatica). Journal of Ecological Engineering 2023, 24, 213–223, doi:10.12911/22998993/158564

           Point 7. To help the reader grasp the study's workflow more easily, the authors could include more visuals to the materials and methods section in the form of figures rather than sticking with the text that now predominates.

Response 7:              

           Point 8. Additional information about tools used, such as the maker, country, and specification, should be included.

Response 8:              

Sonication was done by introducing 3 cm of an ultrasonic horn (22 mm diameter and 100 mm length, S24d22D, Hielscher Ultrasonic, Germany) with 400 W ultrasound equipment (UP400St, Hielscher Ultrasonic, Germany) at 70 % initial amplitude and 23.0 ± 0.9 kHz frequency.

Point 9. The revised manuscript after peer review must provide detailed information on the error and tolerance of the experimental equipment utilized in this study. Due to the disparate outcomes of other researchers' subsequent studies, it would make for a valuable discussion.

Response 9:              

The UP400St Hielscher Ultrasonic equipment was always calibrated before every experiment according to the Germany maker, as well as the analytical balances and the incubator.

Point 10. Results comparison with similar previous studies needs to give.

Response 10:              

The exopolysaccharide concentration of commercial brands of kefir beverages in Russia was investigated. They evaluated seven different kefir beverage brands presenting 50.9 ± 8.1 to 202.5 ± 19.4 mg/L kefiran.

In a previous study, non-fat reconstituted milk was pretreated with HIUS at 22 ± 1.25 kHz, 20 W/cm2, and 90 W/L for 3 min and fermented during 8-12 h with kefir grains. The highest kefiran amount found in the kefir beverage was 204 ± 0.94 mg/L. It was also found that 18% more kefiran in ultrasonicated treatments than in control [29].

Potoroko et al. [29] reported that HIUS pretreatment on reconstituted milk generated exopolysaccharide biosynthesis at the first hours of fermentation time, and it had its peak at 24-60 h in the stationary phase. Cheirsilp and Radchabut [30] found that kefiran production began in the early exponential growth phase when using a mixed culture of Lactobacillus kefiranofaciens JCM 6985 and Saccharomyces cerevisiae IFO 0216

It has been reported that UHT milk at 80 rpm agitation rate and organic skim milk at 125 rpm rotation rate fermented for 24 h at 25 C had 18.8 g/L and 38.9 g/L kefir grains biomass increase, respectively [31,32]. They increased their weight due to the growth of microorganisms and the biosynthesis of their chemical components, such as proteins and exopolysaccharides.

Previous studies reported that 60 g/L deproteinized CW lactose substrate fermented for 120 h at 25 °C, without agitation, and low-fat cow's milk substrate fermented for 24 h at 25 °C, 100 rpm agitation rate had 43.0 % and 32 % kefir grains biomass increase [19,34].

Motaghi et al. [35] fermented cow's milk with kefir grains at 25 °C for 24 h, without agitation. They reported a 1.47 % titratable acidity value. Treatments T4, T3, and T5 also presented the highest kefir grains biomass concentrations at the same fermentation time, 44.2 ± 0.8 g/L, 43.6 ± 0.9 g/L, and 42.9 ± 0.0 g/L, respectively.

It has been reported that during the first 24 h of incubation, homofermentative lactic acid streptococci grow rapidly, causing a drop in pH [8]. Zajšek and Goršek [36] reported a 4.22 ± 0.04 pH value in HTP milk inoculated by kefir grains and incubated at 24 °C for 24 h.

Point 11.  The authors need to improve the discussion in the present article become more comprehensive. The present form was insufficient.

Response 11:              

The discussion was improved to give a better explanation of our findings.

Point 12. Please include the limitation of the present study, it is missing.

Response 12:              

This study is missing a sensory analysis and metabolomic studies to develop a CW kefir accepted by consumers and to understand the HIUS effect on the metabolic pathways during fermentation time.

Point 13. Mention further research in the conclusion section.

Response 13:              

Sensory analysis and metabolomic studies are needed to develop a CW kefir accepted by consumers and to understand the HIUS effect on the metabolic pathways during fermentation time.

Point 14. The authors should enrich their references in the revised manuscript from five years ago. MDPI reference is strongly recommended.

Response 14:              

More MDPI references were added to the manuscript.

Point 15. The manuscript needs to be proofread by the authors since it has grammatical and language issues.

Response 15:              

This manuscript was grammatically improved.

Point 16. After revision, provide a graphical abstract for submission.

Response 16:              

Reviewer 2 Report

Dear authors,

I revised the manuscript about the study kefiran production in ultrasound treated CW kefir. The idea is interesting. The english is Ok.

You can find comments below:

Line 87-88 The sentences were written two times.

Figure 2. y-axis label needs revision such as “kefir grain biomass (g/L)”. Because the increment is given as % increment in Table 3.

Author Response

Dear Prof. reviewer, thank you so much for your comments.

These are our responses:

Point 1. Line 87-88 The sentences were written two times.

Response 1:              

They were inoculated into commercial UHT skimmed cow's milk (48 g/L carbohydrates, 31.2 g/L protein, and 6 g/L fat) and incubated at 25 ± 3 °C for 24 h without stirring. Kefir grains were retrieved by plastic sieving and rinsed with distilled water.

Point 2. Figure 2. y-axis label needs revision such as “kefir grain biomass (g/L)”. Because the increment is given as % increment in Table 3.

Response 2:

Figure 2 was revised. Kefir grain biomass increase was reported first through figure 2 using the units grams / Liter during fermentation time because authors wanted to explore its kinetic. The authors considered it essential to present kefir grain biomass increase as %, too, because it lets us know at what time and treatment presented the highest increments in the experiment. 

Thank you!

Reviewer 3 Report

The authors investigated the effect of high-intensity ultrasound on kefir grains biomass increase and specific metabolites in cheese whey kefir. The treatment of high-intensity ultrasound could enhance the biosynthesis of kefir grains biomass and specific metabolites. This is a routine study. This manuscript was well written. But some problems need to be corrected.

1. Please specify the "+" and "-" in Table 1 in the table notes.

2. Lines 147-148. Why does “kefiran increase” fluctuate at different fermentation times? Please analyze.

3. Line 176. Please correct the unit after "25".

4. The "+" and "-" in Table 4 are too large. Please improve.

Author Response

Dear Prof. reviewer, thank you so much for your comments.

These are our responses:

Point 1. Please specify the "+" and "-" in Table 1 in the table notes.

Response 1:

Data = all treatments mean values comparing with the last value obtained before âž•= the parameter concentration increased comparing with the value obtained before. âž–= the parameter concentration decreased comparing with the value obtained before.

Point 2. Lines 147-148. Why does “kefiran increase” fluctuate at different fermentation times? Please analyze.

Response 2:

Kefiran increase fluctuated at different fermentation times because exopolysaccharide production depends on the growth conditions and the medium chemical composition, especially the absolute quantities, sources, and ratio of carbon/nitrogen. In this experiment, the growth conditions were always the same, but the medium chemical composition changed during fermentation due to biomass increase. Kefir grains microorganisms mainly use kefiran to protect from other microorganisms and as a source of carbon.

Point 3. Line 176. Please correct the unit after "25".

Response 3:

skim milk at 125 rpm rotation rate fermented for 24 h at 25 °C had 18.8 g/L and 38.9 g/L

Point 4. The "+" and "-" in Table 4 are too large. Please improve.

Response 4:

They were improved.

Parameters change during cheese

 whey kefir fermentation

Fermentation time (h)

Parameter

8

16

24

32

Kefir grains biomass (g/L)

âž•11.7

âž–1.4

 âž• 3.0

âž• 3.2

Kefiran (glucose, mg/L)

âž•71.9

âž• 4.7

âž– 15.5

âž• 9.8

Soluble solids (g solut/100 mL solution)

âž– 0.23

âž• 0.18

âž– 0.34

âž– 0.03

Reviewer 4 Report

The manuscript entitled "Effects of high-intensity ultrasound pretreatment on the exopolysaccharide concentration and biomass increase of cheese whey” is a study about a known technology applied to the side-stream of the cheese process production to evaluate the production of the exopolysaccharides of a microbial community such as kefir. This is an exploratory study to identify key factors to take advantage of the cheese whey for kefir fermentation but it has flaws regarding the presentation of the results, data reporting, and the analysis of the experimental design. The statistical analysis of the experimental treatments can be considered as a factorial design 22 (HIUS: 9.0 and 18.0 W/cm2; HIUS time: 30 and 180 s) by fixing the time that enhanced exopolysaccharide and kefir grains production, probably 16 h of fermentation. Concerning the presentation of the results table 2 shows the same experimental results as in figure 1 but converted into percentages, they are not independent results. The same situation with the data shown in table 3 and figure 2. Finally, table 4 indicates the changes in the different parameters measured but does not express to which time of the crop these data correspond. On the other hand, ethanol precipitation shows evidence of total exopolysaccharide production; it is not specific for kefiran (y-axis of figure 1).

Author Response

Dear Prof. reviewer, we really appreciate your relevant comments.

These are our responses:

The manuscript entitled "Effects of high-intensity ultrasound pretreatment on the exopolysaccharide concentration and biomass increase of cheese whey” is a study about a known technology applied to the side-stream of the cheese process production to evaluate the production of the exopolysaccharides of a microbial community such as kefir. This is an exploratory study to identify key factors to take advantage of the cheese whey for kefir fermentation but it has flaws regarding the presentation of the results, data reporting, and the analysis of the experimental design. The statistical analysis of the experimental treatments can be considered as a factorial design 22 (HIUS: 9.0 and 18.0 W/cm2; HIUS time: 30 and 180 s) by fixing the time that enhanced exopolysaccharide and kefir grains production, probably 16 h of fermentation.

Point 1:

Concerning the presentation of the results table 2 shows the same experimental results as in figure 1 but converted into percentages, they are not independent results. The same situation with the data shown in table 3 and figure 2.

Response 1:

            The authors agree with the reviewer´s comments. This study pretends to explore the effects of different intensities and time of high-intensity ultrasound (HIUS) pretreatment on cheese whey fermented with kefir grains.

Potoroko et al., (doi:10.1016/J.ULTSONCH.2018.06.019) presented their exopolysaccharide results as kefiran weight content (mg/L) and as a percentage of increase. On the other side, Dailin et al. (https://www.researchgate.net/publication/278157462_Development_of_cultivation_medium_for_high_yield_kefiran_production_by_Lactobacillus_Kefiranofaciens#fullTextFileContent) reported kefir grains biomass increase as g/L and as a percentage of increase.

Even though data from the tables was obtained from the kinetics of fermentation, both information are relevant. The percentages were obtained considering initial values (0 h of fermentation). For example, figure 1 shows that the highest concentration of total exopolysaccharides was observed in treatment 5 at 16 h of fermentation. Conversely, table 2 shows that treatment 3 presented the highest percentage of the total exopolysaccharide increase at 8 h of fermentation. Both results are important findings. These issues are relevant because now we know that high-intensity ultrasound had different effects on the chemical structure of the cheese whey.

Point 2:

Finally, table 4 indicates the changes in the different parameters measured but does not express to which time of the crop these data correspond.

Response 2:

There are four columns on that table. The columns correspond to 8, 16, 24 and 32 h of fermentation. This data is on the top of those columns. This table describes the relation between several relevant parameters during the fermentation time.

Point 3:

On the other hand, ethanol precipitation shows evidence of total exopolysaccharide production; it is not specific for kefiran (y-axis of figure 1).

Response 3:

Cheirsilp and Radchabut (doi:10.1016/J.NBT.2011.01.009) used ethanol precipitation as an analytical methodology to quantify kefiran. However, the authors agree with the reviewer because it may be more exopolysaccharides biosynthesized by the complexity of the kefir microorganism’s consortium. So, we agree to change kefiran for total exopolysaccharides. Thank you so much for your recommendation.

The manuscript was also improved.

Thank you so much for your comments.

Reviewer 5 Report

This paper describes the effect of high-intensity ultrasound pretreatment on some general properties of kefir made with cheese whey. As such, the manuscript is interesting, well-presented and clear and provides information that will improve future efforts on kefir research.

Still, there are some minor comments and corrections about the document:

-Line 23: please use lower case for "Total"

-Line 23, 25, 39, 145, 194, 199, 241: I have the feeling that you need to include a semicolon or a parenthesis when refering to the results of two treatments. For example, "Ultrasonicated CW at 18 W/cm2 for 30 and 180 s at 24 h fermentation time had significantly higher kefir grains biomass (p < 0.05) than the control; 44.2 ± 0.8 25 and 43.6 ± 0.9 g/L, and 40.5 ± 0.4 g/L, respectively. Please check throughout the text for similar sentences.

-Line 57: refer to "microbiota" instead "microflora".

-Line 78: "Experimental flow diagram....."

-Line 146: use lowercase for "The"

-Line 163: Please try to explain why your study resulted in higher exopolysaccharide than reference 34.

-Line 204: Improve the writting in the sentence. "These microorganisms were lactobacilli, lactococci, and yeasts with ability to auto-aggregate, co-aggregate, form biofilm, and properties such as hydrophobicity and cell surface. What do you mean at the end of this sentence with "cell surface"?

-Line 209: Improve writting. Wang et al. also found that.......

-Line 210: Regarding this observations, consider the possibility that different microbial populations are growing at different fermentation times. This may also explain the diauxic increase in biomass.

-Line 251: Write "decrease"

-Line 260: there is an extra "pH"

-Line 285: Not clear if the values are the average of all treatments. Please clarify.

Author Response

Dear Prof. reviewer, thank you so much for your comments.

These are our responses:

Point 1: Line 23: please use lower case for "Total"

Response 1:              

CW pretreated with 18.0 ± 3.0 W/cm2 for 180 s and fermented for 16 h had significantly higher (p < 0.05) total exopolysaccharide concentration

Point 2: Line 23, 25, 39, 145, 194, 199, 241: I have the feeling that you need to include a semicolon or a parenthesis when refering to the results of two treatments. For example, "Ultrasonicated CW at 18 W/cm2 for 30 and 180 s at 24 h fermentation time had significantly higher kefir grains biomass (< 0.05) than the control; 44.2 ± 0.8 25 and 43.6 ± 0.9 g/L, and 40.5 ± 0.4 g/L, respectively. Please check throughout the text for similar sentences.

Response 2:              

Line 23 exopolysaccharide concentration than the control; 212.7 ± 0.0 and 186.6 ± 0.0 mg/L, respectively.

Line 25 Ultrasonicated CW at 18 W/cm2 for 30 and 180 s at 24 h fermentation time had significantly higher kefir grains biomass (p < 0.05) than the control; 44.2 ± 0.8 and 43.6 ± 0.9 g/L, and 40.5 ± 0.4 g/L, respectively.

Line 39 oxygen demand (COD) and biological oxygen demand (BOD); 50-102 g/L and 27-60 g/L, respectively.

Line 145 concentration than control; 212.7 ± 0.0 and 186.6 ± 0.0 mg/L, respectively.

Line 194 control; 44.2 ± 0.8 and 43.6 ± 0.9 g/L, and 40.5 ± 0.4 g/L, respectively.

Line 199 pretreatment at 18 W/cm2 for 30 s presented the highest values; 46.9 ± 0.5 and 46.9 ± 1.6.

Line 241 rom CW kefir; 1.08 ± 0.01 %, 1.07 ± 0.02 %, 1.06 ± 0.03 %, and 1.01 ± 0.01 %, respectively.

Point 3: Line 57: refer to "microbiota" instead "microflora".

Response 3:                          

presence of 522 species in the microbiota of this multi-functional beverage.

Point 4: Line 78: "Experimental flow diagram....."

Response 4: Experimental flow diagram to study

Point 5: Line 146: use lowercase for "The"

Response 5: In a previous study, the exopolysaccharide concentration of commercial brands of kefir beverages.             

Point 6: Line 163: Please try to explain why your study resulted in higher exopolysaccharide than reference 34.

Response 6:

            Potoroko et al. [34]  pretreated non-fat reconstituted milk with HIUS at 22 ± 1.25 kHz, 20 W/cm2, and 90 W/L for 3 min and fermented during 8-12 h with kefir grains. The exopolysaccharide amount reported in the kefir beverage was 204 ± 0.94 mg/L. On the other side, we pretreated fresh CW with 18 W/cm2 for 3 min and fermented for 16 h. In this study we found 212.7 ± 0.0 mg/L EPS. The HIUS pretreatments were very similar, however we fermented CW for 4 more hours. This fact may explain the EPS increase difference.

           Point 7: Line 204: Improve the writting in the sentence. "These microorganisms were lactobacillilactococci, and yeasts with ability to auto-aggregate, co-aggregate, form biofilm, and properties such as hydrophobicity and cell surface. What do you mean at the end of this sentence with "cell surface"?

Response 7:  According to Wang et al. the formation of grains in kefir has been associated with the cell surface properties of kefir grains microorganisms. For example, most of the lactic acid bacteria strains evaluated in that study had a negative charge on their cell surface. Therefore, they proposed that grain formation begins with the self-aggregation of Lactobacillus kefiranofaciens and Saccharomyces turicensis to form small granules.

           Point 8: Line 209: Improve writting. Wang et al. also found that.......

Response 8: Wang et al. [38] also found that kefir grains biomass

Point 9: Line 210: Regarding these observations, consider the possibility that different microbial populations are growing at different fermentation times. This may also explain the diauxic increase in biomass.

Response 9:              

Point 10: Line 251: Write "decrease"

Response 10: All the HIUS CW treatments showed a decrease in pH through fermentation time.

Point 11: Line 260: there is an extra "pH"

Response 11: A previous study established that pH stabilizes at 3.3 (it was erased).

Point 12: Line 285: Not clear if the values are the average of all treatments. Please clarify.

Response 12: This part of the study was very exciting. The results showed clear tendencies through fermentation time. It was observed that the maximum formation of kefir grains biomass, and the total EPS was at 8 h fermentation also. The values of this table corresponded to the average of all treatments at each fermentation time.

References

Potoroko, I.; Kalinina, I.; Botvinnikova, V.; Krasulya, O.; Fatkullin, R.; Bagale, U.; Sonawane, S.H. Ultrasound Effects Based on Simulation of Milk Processing Properties. Ultrason Sonochem 2018, 48, 463–472, doi:10.1016/J.ULTSONCH.2018.06.019.

Wang, S.Y.; Chen, K.N.; Lo, Y.M.; Chiang, M.L.; Chen, H.C.; Liu, J.R.; Chen, M.J. Investigation of Microorganisms Involved in Biosynthesis of the Kefir Grain. Food Microbiol 2012, 32, 274–285, doi:10.1016/j.fm.2012.07.001.

Thank you so much for your comments!

Reviewer 6 Report

The manuscript entitled "Effects of high-intensity ultrasound pretreatment on the exopolysaccharide concentration and biomass increase of cheese whey kefir" was reviewed. The reviewed version seems to be a revised version of a manuscript. In this study, the authors have attempted to apply high intensity ultrasound pretreatment to increase the content of Kefiran grain   and exopolysaccharides in cheese whey-based Kefir.

The manuscript writing had been edited well, except for some minor corrections. The overall topic of this study fits with the journal scope, and the manuscript has enough novelty as the authors have used cheese whey as the substrate for kefir production. However, it lacks some essential experiments such as microbial analyses which are important for such products. Some of the observations addressed by the authors may be related to the changes in microbial differences in different treatments. Thus, the authors are recommended to improve this study by adding the results of microbial analyses (such as Bacteria: yeasts ratio), if possible. Moreover, some metabolite analysis (organic acids and ethanol content) could improve the study.

In addition to the above comments, the authors are recommended to consider the comments mentioned below:

Abstract

L-27: A short overall conclusion is needed at the end of Abstract.

Introduction

L-53-54: It is not clear. Please rewrite the sentences to make it more understandable.

L-72-73: As you noted, it is very interesting to analyze the microbial metabolites under the effect of US pretreatment.  But the study lacks such data. Please improve the manuscript by adding data from lactic acid and ethanol contents (if available).   

Materials and Methods

That is OK.

Results and discussion

Data in Table 3, and Figure 2 can be merged.

Fig 1, and L-142: There is a high variation in EPS content (kefir grains) at time 0 for different treatments. Was it not possible to minimize the variations at time 0? Some differences observed during fermentation may be resulted from variations in kefir grain content at time 0.

L-146: "The" should be replaced with "the"

L-183, Table 2. How did you calculate EPS percentage? Please note as clear that the EPS increase percentage was calculated according to the difference among concentrations at time 0 and those at other time intervals.    

L-225-226: Please check the findings with that expressed in Abstract (L-24-26). Were the differences significant?                         

Author Response

Dear Prof. reviewer, thank you so much for your comments.

These are our responses:

Point 1:

The manuscript writing had been edited well, except for some minor corrections. The overall topic of this study fits with the journal scope, and the manuscript has enough novelty as the authors have used cheese whey as the substrate for kefir production. However, it lacks some essential experiments such as microbial analyses which are important for such products. Some of the observations addressed by the authors may be related to the changes in microbial differences in different treatments. Thus, the authors are recommended to improve this study by adding the results of microbial analyses (such as Bacteria: yeasts ratio), if possible. Moreover, some metabolite analysis (organic acids and ethanol content) could improve the study.

Response 1:              

First of all, we would like to thank you for your excellent comments. This research was an exploratory study. Initially, we planned to obtain the kinetics of lactose metabolism and metabolites such as organic acids (acetic acid, pyruvic acid, citric acid and hippuric acid), and ethanol. Also, we planned to measure the degree of proteolysis through the fermentation time. However, the COVID-19 pandemic times came, so we had to stop everything in Mexico. The project´s financial support was closed. So, we had to focus on the main objectives. This research shows that it is possible to increase kefiran production and kefir grains biomass using fresh cheese whey pretreated with high-intensity ultrasound, transforming a potent environmental pollutant into a valuable substrate.

Point 2: L-27: A short overall conclusion is needed at the end of Abstract.

Response 2:              

Fresh CW pretreated with HIUS enhanced the biosynthesis of kefir beverage total exopolysaccharides concentration and kefir grains biomass.

Point 3: L-53-54: It is not clear. Please rewrite the sentences to make it more understandable.

Response 3:                          

Biomass produced by microorganisms and vegetables usually presents nontoxicity and versatility, so it has been evaluated in recent research as raw material through industrial encapsulating processes

Point 4: L-72-73: As you noted, it is very interesting to analyze the microbial metabolites under the effect of US pretreatment.  But the study lacks such data. Please improve the manuscript by adding data from lactic acid and ethanol contents (if available).   

Response 4:              

We did not have enough time to get that information because of the COVID-19 pandemic. Besides, the project´s financial support ended up.

Point 5: Data in Table 3, and Figure 2 can be merged.

Response 5:              

Data in Table 3 and Figure 2 show that kefir grains biomass increase during fermentation. Figure 3 presents the kefir grains biomass kinetic using g/L units and its statical analysis comparing all treatments at the same fermentation time. Conversely, Table 3 shows kefir grain biomass increase as %. The statistical analysis compared kefir grains biomass of all treatments at the same fermentation time and all fermentation times at each treatment. It is essential to mention that kefir grains biomass increase can be reported using g/L and % units. We used both to have a better understanding of the results.

Point 6: Fig 1, and L-142: There is a high variation in EPS content (kefir grains) at time 0 for different treatments. Was it not possible to minimize the variations at time 0? Some differences observed during fermentation may be resulted from variations in kefir grain content at time 0

Response 6:              

            This is an exciting issue for us, too. Even though there was a high variation in total exopolysaccharides content, the results were not statistically different (p < 0.05) at this fermentation time. However, we repeated this part twice. It is essential to say that kefir grains were inoculated into non-fat milk more than 30 times every two days before this experiment. Hence, the metabolism of kefir microorganisms was very active. Another important point was that the highest EPS concentration values were observed when fresh CW was pretreated with the highest HIUS (18 W/cm2). Zhou et al. (2020) applied HIUS (20-40 % amplitude for 1-10 min) to goat milk b-Lactoglobulin. They reported that the percentage of b-sheet was significantly decreased while those of a-helix and random coils increased.

Moreover, their results showed that the surface hydrophobicity index and intrinsic fluorescence intensity of samples were enhanced and increased with increasing HIUS amplitude or time. In this study, fresh CW samples were pretreated at 70% amplitude. Total EPS results showed that at 0 h fermentation time fresh CW samples pretreated at 18 W/cm2 for 30 s and 180 s had 136.3 ± 0.0 and XX glucose mg/L, respectively, meanwhile, samples pretreated at 9 W/cm2 for 30 s and 180 s had 108.5 ± 0.0 and XX glucose mg/L, respectively. Therefore, we have hypothesized that the highest HIUS pretreatments may modify secondary and tertiary milk protein's chemical structures, enhancing the enzymatic activity of kefir grains microorganisms to stimulate the biosynthesis of total EPS.  

           Point 7: L-146: "The" should be replaced with "the"

Response 7:      

In a previous study, the exopolysaccharide concentration of commercial brands of kefir beverages in Russia was investigated        

           Point 8: L-183, Table 2. How did you calculate EPS percentage? Please note as clear that the EPS increase percentage was calculated according to the difference among concentrations at time 0 and those at other time intervals

Response 8:              

The total EPS percentage was calculated at every fermentation time using the following equation:

Total EPS (%) = [Glucose]tx – [Glucose]t0   x  100

                                    [Glucose]t0

tx = glucose concentration at a specific fermentation time at each treatment

t0 = glucose concentration at 0 h fermentation time at each treatment

Point 9: L-225-226: Please check the findings with that expressed in Abstract (L-24-26). Were the differences significant?

Response 9:              

Kefir grains biomass increase reported in Line-225-225 was obtained at 32 h fermentation time. So, it was modified “The highest biomass increase values were 53.6 – 56.4 % at 32 h fermentation. At this fermentation time, it was not found a significant (p > 0.05) difference between control and CW ultrasonicated treatments”. The findings expressed in the abstract (L-24-26) corresponded to kefir grains biomass increase obtained at 24 h fermentation time. They were statistically different (p < 0.05).

Thank you so much for your comments!

References

Zhou et al. (2020). Effects of high intensity ultrasound on physiochemical and structural properties of goat milk B-lactoglobulin. Molecules, 25(16), 3637.

Thank you so much for your valuable comments!

Round 2

Reviewer 1 Report

Accept in the present form for this manuscript.

Author Response

Thank you for your kind comments.

Reviewer 2 Report

Dear authors,

In my previous review circle, some of my comments have not delivered to you. Here you can see my other comments on the manuscript and study.

I revised the manuscript about the study on ultrasound treatment effect on kefiran production in CW. Although the idea is interesting, the study construction is very poor.

The ultrasound conditions almost provided no significant effect on the results. To maximum and minimum points of treatment conditions should have choosen to see the effect of Ultrasound amplitudes and application time.

The parameters are not suitable to differentiate the true effect of ultrasound application on kefiran production.

The sample characterisation is also very primitive.   

Author Response

Dear Prof. reviewer, we really appreciate your relevant comments.

These are our responses:

Point 1. I revised the manuscript about the study on ultrasound treatment effect on kefiran production in CW. Although the idea is interesting, the study construction is very poor.

Response 1:

The experiment was statistically designed as a factorial 22 (High-intensity ultrasound: 9.0 ± 2.7 and 18 ± 3.0 W/cm2, and ultrasonication time: 30 and 180 s). The control was also considered. Every run, we manufactured Fresco-style cheese using the same commercial pasteurized low-fat milk and following the same methodology to get fresh cheese whey. So, once we got it, we cooled it at 6 °C, ultrasonicated and inoculated it with kefir grains to start the fermentation process on the same day. Simulating normal activities at cheese-making facilities was very important for the experiment to get fresh cheese whey. The cheese whey was fermented for 40 h, and samples were obtained at 0, 8, 16, 24, 32, and 40 h. The five treatments were done in triplicate, at least. Each sample was analyzed through kefiran production, biomass increase, ° Brix, titratable acidity, and pH. A robust statistical analysis of data was done using ANOVA (a= 0.05) and the Tuckey-Kramer test (p < 0.05). The kinetic of each parameter was obtained. Besides, kefiran production results were analyzed and reported as glucose mg/L and %; meanwhile, kefir grains biomass increase was presented as g/L and %. These results are reported for the first time. This was an exploratory study.

Now we know that it is possible to increase kefiran production and kefir grains biomass using cheese whey pretreated with high-intensity ultrasound, transforming a potent environmental pollutant into a valuable substrate. Results demonstrated that cheese whey ultrasonicated at 18 ± 3.0 W/cm2 for 180 s and fermented by kefir grains for 16 h presented the highest kefiran production. These results suggest that higher ultrasonication intensity and time may be related to higher kefiran biosynthesis. Kefiran is kefir grains microorganism’s biopolymer metabolite with several uses in the food and pharmaceutical industries. High-intensity ultrasound had similar effects on the kefir grains biomass increase. Cheese whey ultrasonicated at 18 ± 3.0 W/cm2 also presented the highest values. Until now, there is a hypothesis of the mechanism of kefir grains formation through a model using specific microorganisms previously isolated from kefir grains. These results support it from a realistic point of view because the whole kefir grains microorganisms were inoculated, and their behavior growth curve matches the hypothesis. 

Point 2: The ultrasound conditions almost provided no significant effect on the results. To maximum and minimum points of treatment conditions should have choosen to see the effect of Ultrasound amplitudes and application time.

The parameters are not suitable to differentiate the true effect of ultrasound application on kefiran production.

The sample characterisation is also very primitive.  

Response 2:

The authors agree with the reviewer´s comments. We also expected that all high-intensity pretreatments results would have been statistically significant. We started with this exploratory study to determine key factors to take advantage of this emergent technology and the possibility of using cheese whey as a substrate. Cheese whey pretreated with 18 ± 3.0 W/cm2 high-intensity ultrasound (HIUS) for 180 s and fermented for 16 h presented significantly higher (p < 0.05) kefiran concentration than control, 212.7 ± 0.0 and 186.6 ± 0.0, respectively. Even though this was the only significant difference, cheese whey pretreated with HIUS treatments showed a higher kefiran production tendency most of the fermentation time (Figure 1). Similar results were observed at kefir grains biomass increase. Cheese whey pretreated with 18 ± 3.0 W/cm2 HIUS for 30 and 180 s and fermented for 24 h presented significantly higher (p < 0.05) biomass increase than control, 44.2 ± 0.8 and 43.6 ± 0.9 g/L, and 40.5 ± 0.4 g/L, respectively. Figure 2 shows that cheese whey pretreated with HIUS had a higher kefir grains biomass increase than control during fermentation time. 

The manuscript was also improved.

Thank you so much for your comments.

Reviewer 4 Report

Some changes were included and the manuscript was improved.

Author Response

Thank you so much for your valuable comments!

Reviewer 6 Report

The authors have improved the manuscript as far as they could, and therefore it can be accepted.

Round 3

Reviewer 2 Report

Dear authors,

Thanks for your answers and explanations to my comments. Althogh a huge work load was done in the study, as I stated before the sudy does not have remarkable outputs. The study focused on effect of ultrasound application however this application provided significant effect on the sample as expected, the sufficient characterization studies were not done. We can not see how the US affected the sample. The structural change, solubility change, etc? Moreover the ultrasound parameters were not broad enough to see the ultrosound effect on the final sample. In the second part, fermentation, we can see the fermentetion process almost provided the same effect on kefir grain production at certain times. There were almost no differences between different traits. The results seems like that similar results could be also obtained by applying a random US treatment and fermentation process. Both the ultrasund treated sample and fermentation process were not characterized well so that, the result discussions and data evaluations were at very basic level. I still believe that the manuscript with this quality is not well enough to be published in Processes journal. 

Author Response

(The authors gave the same response as above.)
